# An Exploratory Study of the Potential of Online Counseling for University Students by a Human-Operated Avatar Counselor

**DOI:** 10.3390/healthcare12131287

**Published:** 2024-06-27

**Authors:** Keita Kiuchi, Hidehiro Umehara, Koushi Irizawa, Xin Kang, Masahito Nakataki, Minoru Yoshida, Shusuke Numata, Kazuyuki Matsumoto

**Affiliations:** 1Japan National Institute of Occupational Safety and Health, Japan Organization of Occupational Health and Safety, Kawasaki 214-8585, Japan; 2Graduate School of Biomedical Sciences, Department of Psychiatry, Tokushima University, Tokushima 770-0042, Japan; umehara.hidehiro@tokushima-u.ac.jp (H.U.); irizawa.koushi@tokushima-u.ac.jp (K.I.); nktk@tokushima-u.ac.jp (M.N.); shu-numata@tokushima-u.ac.jp (S.N.); 3Graduate School of Technology, Industrial and Social Sciences, Tokushima University, Tokushima 770-8506, Japan; kang-xin@is.tokushima-u.ac.jp (X.K.); mino@is.tokushima-u.ac.jp (M.Y.); matumoto@is.tokushima-u.ac.jp (K.M.)

**Keywords:** avatar counseling, human-operated avatars, university students, digital mental health, empathy, communication, anthropomorphism, animacy, likeability, perceived intelligence

## Abstract

Recently, the use of digital technologies, such as avatars and virtual reality, has been increasingly explored to address university students’ mental health issues. However, there is limited research on the advantages and disadvantages of counselors using avatars in online video counseling. Herein, 25 university students were enrolled in a pilot online counseling session with a human counselor-controlled avatar, and asked about their emotional experiences and impressions of the avatar and to provide qualitative feedback on their communication experience. Positive emotions during the session were associated with impressions of the avatar’s intelligence and likeability. The anthropomorphism, animacy, likeability, and intelligent impressions of the avatar were interrelated, indicating that the avatar’s smile and the counselor’s expertise in empathy and approval may have contributed to these impressions. However, no associations were observed between participant experiences and their prior communication with avatars, or between participant experiences and their gender or the perceived gender of the avatar. Accordingly, recommendations for future practice and research are provided. Accumulating practical and empirical findings on the effectiveness of human-operated avatar counselors is crucial for addressing university students’ mental health issues.

## 1. Introduction

The use of digital technologies in addressing the mental health issues of university students is advancing. For example, avatar-led digital interventions for smoking cessation have demonstrated high satisfaction levels in the involved participants [1]. Meanwhile, improvements in anxiety and depression through digital health interventions incorporating gamification [2] and improved effective of cannabis education [3] have also been reported. For supporting the mental health of university students with autism spectrum disorder (ASD), computer-assisted programs are implemented along with mentoring programs through support groups [4]. In Canadian universities, the utilization of interactive technologies is being considered to increase the accessibility of mental health support for international students [5]. Regarding apps used for maintaining mental well-being and managing drug usage, co-development with young users, including that of graphical interfaces, is being promoted [6]. In the digital mental health management domain, effectively utilizing avatars is crucial for increasing user accessibility and commitment.

### 1.1. Literature Review

#### 1.1.1. Definition of Avatars and Their Usage Patterns

Basically, avatars are visually represented digital expressions in cyberspace [7]. However, in a broader sense, avatars encompass all types of digital representations, including voice [8]. In the field of robotics, entities that perform various tasks on the behalf of humans are known as robotic avatars or avatar robots [9]. This study primarily focuses on visually represented avatars on two-dimensional (2D) video conferencing screens while comprehensively considering results related to various avatars, including robots.

Conventional utilization of avatars in the mental health domain can be categorized into the following four groups: first, as a development of e-mental health, initiatives such as adding avatars to smartphone apps or utilizing autonomous computer agents have been reported [10,11]. Second, psychotherapy based on virtual reality (VR) has been implemented [12]. Third, mental health support provided using robots offers insights regarding avatars [13]. Finally, in the online counseling domain, there are instances where the client and/or counselor conduct sessions using avatars [14,15,16].

This study involves an exploratory investigation of counselors operating avatars in the context of online counseling. The term “AVATAR therapy” sometimes refers to a method that promotes a patient’s coping with auditory hallucinations through a three-way relationship between the therapist, the client, and an avatar—a digital embodiment of the client’s mental image of the hallucinated voice’s owner [17]. Herein, we consider “avatar counseling” as a format wherein a human-counselor-operated avatar and a client conduct sessions of general online counseling rather than AVATAR therapy. When referring to “avatar-based” counseling, there are methods that focus on the “client as avatar” [14,15]. In such methods, the client’s selection or creation of specific avatars and experiences of interacting with a virtual world are therapeutically utilized. The counselor does not necessarily need to be embodied as an avatar. The “client as avatar” aspect is outside the scope of this study.

#### 1.1.2. Knowledge of the Effectiveness of Avatar Counseling

The effectiveness of avatar-based interventions has been demonstrated for various mental health conditions. One of the disorders that can particularly benefit from the use of avatars is ASD. For ASD, a case has been reported in which a client with comorbid ASD and social anxiety disorder (SAD) was able to talk to a counselor through a robot as an avatar [18]. In addition, the effectiveness of social skills lessons using live-animation avatars for adolescents with ASD [19] and increased response, conversation time, and attention span in children using avatar robots [20] have been recognized. In the case of ASD, communication with avatars may be more comfortable than that with humans. For example, infants with ASD responded more frequently to simple humanoid robots and simple avatars than to humans [21]. Moreover, children with ASD showed reduced excessive attention to eyes and increased interest in facial expressions and emotional expressions when communicating with avatars [22].

SAD and generalized anxiety disorder (GAD) are disorders for which the effectiveness of avatar-based therapy can be expected. To date, the effectiveness of VR-based psychotherapy for SAD and GAD has been clarified to a certain extent [23,24]. However, according to a review study, further verification of the effectiveness of VR therapy for SAD is necessary [25]. Moreover, the use of avatars has not received much attention.

In addition, avatar-based interventions are being applied to psychotherapy for various chronic diseases as a new method in digital mental health [26]. Particularly for depression, the effectiveness of dialogue-based self-management intervention apps [27] and cognitive behavioral therapy apps that allow users to customize their own avatars [28] has been verified. Avatar-based assessment and management tools have also been developed for suicide prevention [29].

The influence of user personality traits on the effectiveness, acceptability, and experience of avatar use has been examined in studies on robots and computer agents. A review of robot-assisted healthcare revealed associations between patient personality and treatment outcomes, acceptability, social/emotional outcomes, and anthropomorphism [30]. Another review showed that user characteristics influence human responses to robots, such as extroverted individuals preferring interaction with robots more than introverted individuals do [31]. Similar associations have been recognized in a review of computer agents [32].

The gender of avatars and robots and user preferences for avatar behavior based on user gender have also been examined. For example, in a study on science, technology, engineering, and math education targeting teenage students, female agents promoted interest and math performance by increasing self-efficacy in both male and female students compared with a control group [33]. Meanwhile, teenage students preferred androgynous avatars, but being completely genderless was not necessarily ideal; it was important for users to be able to assign their own perceived gender to the avatar based on their own judgment [34]. Moreover, customers preferred avatars of a different gender than their own [35]; men preferred avatar behaviors that showed gratitude or offered assistance, whereas women preferred expressions of thanks [36]. Thus, the gender of avatars, user perceptions of avatar gender, and the user’s gender are thought to influence user impressions of communication with avatars.

Several studies on avatar impressions related to trust have been conducted. Five factors that influence trust in agents are the agent’s social intelligence, voice characteristics and communication style, agent appearance, nonverbal communication, and quality of performance [37]. In fact, agents with expertise had higher trust and promoted learning outcomes [38]. However, user characteristics also seem to be related to avatar impressions. For instance, in the context of virtual home tours, real estate agents who engaged in small talk increased trust in extroverted users but not in introverted users, as the latter tended to shy away from small talk, avoid social situations, and prefer serious conversations [37].

Thus, the use of avatars is expected to be effective, particularly for chronic diseases, including ASD, SAD, GAD, depression, and suicidal ideation. In this context, user personality, gender, and the perceived gender of avatars seem to be involved in their effects and impressions.

#### 1.1.3. Research Gaps and Purpose of the Study

Regarding communication between avatars, research has been conducted for some time in Second Life, a precursor to multiuser virtual environments now also known as the metaverse. Attempts at cross-country therapy have already been reported, but the focus is mainly on considerations related to online counseling, which has become common [39]. Meanwhile, regarding interactions between avatars, the use of gaze and the transmission of emotions are reportedly possible [40,41]. The advantage of avatar-mediated communication is that it provides a higher sense of presence compared with communication using only text or voice [42].

Meanwhile, research on communication with avatars using video conferencing systems is limited. In the context of research on computer-mediated communication, perceived intimateness, copresence, and emotionally-based trust do not significantly differ between interactions via voice, video, and avatars in collaborative work through video conferencing systems [43]. Moreover, in the context of online conversations, conversations with avatars wearing white coats are reportedly preferred for serious consultations [44] and that participants reflect more positively on the conversation when conversations are conducted using avatars with emphasized smiles [45].

In a study using the Wizard-of-Oz method (i.e., users are unaware that the avatar is operated by a human) and in situations where users converse with an avatar via a videoconferencing system, through multiple conversations, users perceived avatars as caring partners [46]. Similarly, in another study using the Wizard-of-Oz method, after conversing with an avatar, participants felt better, the intensity of negative emotions decreased, and emotions improved overall [47].

As described above, there are high potential expectations for the use of avatars in digital mental health, and their effectiveness is gradually being recognized in ASD, SAD, GAD, depression, and suicide prevention. However, research findings are mixed, with initiatives in different contexts such as VR, robots, and the metaverse. Therefore, this study aims to exploratively confirm the following three points by experimentally setting up a dialogue scene with an avatar operated by a human counselor using a 2D video conferencing system for university students:(1)The association between the tendencies of ASD, GAD, SAD, depression, suicidal ideation, and experiences in interviews with avatar counselors.(2)The association between participant personality and experiences in interviews with avatar counselors.(3)The association between actual gender, perceived avatar gender, the degree of agreement between actual gender and perceived avatar gender, and experiences in interviews with avatar counselors.

By achieving these objectives, this study is expected to achieve new insights regarding effective methods of counseling using avatar-mediated video conferencing and ways to utilize avatars according to the characteristics of target users.

## 2. Materials and Methods

### 2.1. Research Design

This study involved university students engaging in a conversation with an avatar operated by a counselor for ~30 min and answering questionnaires before and after the conversation. Participants were asked to prepare three topics for the conversation with the avatar in advance: something that made them happy, something that made them sad, and something that made them angry. These three emotions are often examined in research on emotions [48,49,50]. Herein, we used topics that were closer to the counseling context than simple everyday conversations. In addition, unlike problems and goals, these are the topics that all college students can talk about. During the interview, the counselor avatar introduced itself and then asked about these three points in order, listening actively and supportively. The conversation was conducted in a semi-structured interview format. In other words, for each event, the counselor used preprepared questions about the details of the event (when, where, who, and what), the circumstances of its occurrence, the reason for experiencing such an emotion, the scaling of the emotion, thoughts at the time of the event, behaviors at the time of the event, differences between that time and the usual times, reflections on the event, and the possibility of different outcomes or choices. Moreover, it asked additional questions and developed the conversation according to the flow of the conversation.

### 2.2. Participants

The participants were 25 students (6 women and 19 men, average age 22.68 years) recruited from the medical and science and engineering faculties of one national university in Japan. The recruitment was conducted on a first-come, first-served basis, with 10 slots available in the School of Medicine and 15 in the School of Science and Engineering. The exclusion criterion was that the participants not be in sufficient mental and physical health to understand and endure the experiments. However, all participants could attend school without any problems, and no one who met the exclusion criterion applied.

This was an exploratory study, and no prior sample size design was conducted. Participants were recruited as much as possible within the budget.

### 2.3. Counselor Avatar and Counselors

The avatar used was a commercially available 2D avatar called “Icon-style Businessman” [51] (Figure 1). Based on the findings of existing studies, we decided that a human-like, not too realistic, neutral gender avatar with a clear smile expression would be effective. The avatar’s functions included expressing nine preset expressions based on facial tracking: neutral, smile, shyness, pallor, anger, glitter, sweat, surprise, and subtle. Tracking was only for facial expressions, head, and face orientation, and there was no tracking for arms, legs, or torso. There were preset motions, but they were not used in this study. Describing the avatar based on existing classifications [52], its name was explicitly stated as Haru, whereas its gender, race, and age were not specified. Its appearance was an anime-style modification with black straight hair common among Japanese people, eyes with a white circle drawn inside a black circle, and skin color close to white but slightly orange, leaning toward male but with an androgynous appearance. VTube Studio [53] was used for tracking and avatar operation. The overall experimental system and the participant’s view are shown in Figure 2 and Figure 3.

The avatar was operated by six individuals, including the first author. All of them were professionals with a certain level of counseling experience and hold qualifications as psychotherapists in Japan. The first author was male, whereas the other individuals were female. Each of them recruited participants from students in departments they had connections with through collaborative research or hospital work. The person in charge of a department set up the experimental environment at the university from where participants accessed the web conference. Thus, the first author was in charge of participants from technology majors, whereas the other five individuals were in charge of participants from medical majors.

Zoom 5.14.5 [54] was used as the video conferencing system. The avatar image displayed in VTube Studio 1.28.0 was captured into OBS Studio 30.0 [55] and used as a camera image for Zoom (Figure 2). To reduce the influence of the human counselor manipulating the avatar, the voice was converted using MagicMic 5.4.0 [56]. We looked for a voice that was not too high, not too low, pitched, and gender neutral. As a result, a female voice named “Elder Woman” was used.

### 2.4. Materials

In the presurvey, gender, age, major, experience with avatar-based communication, Big Five personality traits, autistic tendencies, generalized anxiety tendencies, social anxiety tendencies, depressive symptoms, and suicidal ideation were investigated. For experience with avatar-based communication, participants were asked to indicate the frequency of “conversations or streaming in which they acted as an avatar or character”, “conversations in which the other person acted as an avatar or character”, and “viewing of streams in which someone acted as an avatar or character” by selecting the options “rarely”, “a few times”, “about once a month”, “about once or twice a week”, and “more than three times a week”.

Big Five personality traits were assessed using the Ten-Item Personality Inventory (TIPI). TIPI-J assesses each element of the Big Five personality traits (extraversion, agreeableness, conscientiousness, neuroticism, and openness) using two seven-point scale items and scores them from 2 to 14 points [57]. Autistic tendencies were assessed using the autism spectrum quotient (AQ). AQ assesses five elements (social skill, attention switching, attention to detail, communication, and imagination) using 10 4-point scale items for each element and scores them from 0 to 10 points [58]. The cutoff point for the total score is ≥33 points [58].

GAD symptoms were assessed using GAD-7, which is a seven-item four-point scale that provides scores from 0 to 21 points, with 0–4 points indicating no symptoms, 5–9 points indicating mild symptoms, 10–14 points indicating moderate symptoms, and 15–21 points indicating severe symptoms [59].

SAD symptoms were assessed using the Liebowitz Social Anxiety Scale (LSAS), which is a 24-item four-point scale (0–3 points) that scores social anxiety based on the total points of fear/anxiety and avoidance in situations related to performance (13 items) and social interaction (11 items) [60]. For performance, the range is 0–39 points each for fear/anxiety and avoidance, with a maximum of 78 points. For social interaction, the range is 0–33 points each for fear/anxiety and avoidance, with a maximum of 66 points, and the maximum for all items is 144 points. Scores of 30–49 points are considered borderline and those of 50–70 points are considered moderate [60].

Depressive symptoms were assessed using the Japanese version of the Patient Health Questionnaire-9 (PHQ-9), which is a nine-item four-point scale that provides scores from 0 to 27 points, with ≥10 points indicating the possibility of depression [61]. Suicidal ideation was assessed using the short form of the Suicidal Ideation Scale (SIS), which was developed by translating the Scale for Suicide Ideation into Japanese, modifying it into a self-report format, and removing specific words indicating suicidal methods to obtain a scale that can be widely implemented in epidemiological surveys [62]. SIS asks participants to respond to six questions regarding the intensity of suicidal ideation, ambivalent feelings, thoughts of suicide attempt, duration of suicidal ideation, frequency of suicidal ideation, and suicide planning using a three-point scale (0–2 points), with a maximum score of 12 points. The mean ± standard deviation of 2486 participants in their 20s to 60s registered with a survey company was reported to be 2.06 ± 2.7 [62].

In the postsurvey, participants were asked to rate their mood during the interview considering four aspects: “happiness”, “anxiety”, “discomfort”, and “sadness”, each on a scale of 0–10 points. In addition, the impression of the avatar was assessed using the Godspeed questionnaire, which is a self-report scale that asks participants to rate their impressions of robots on five-point scales (1–5 points) for five questions on anthropomorphism, six questions on animacy, five questions on likeability, five questions on perceived intelligence, and three questions on perceived safety [63,64]. It has been translated into 19 languages, including Japanese, and is widely used in the field of human–robot interaction [63,64]. It has also been used to evaluate embodied conversational agents [65,66,67] and is suitable for evaluating the impression of human-operated avatars in this study. Regarding perceived safety, as the standard aggregation method was found to have insufficient internal consistency [68], we used a different scoring method. The items “relaxed”, “calm”, and “still” were scored as five points, corresponding to one point for “anxious”, “agitated”, and “surprised”, respectively. This scoring method ensures that higher scores consistently reflect a greater sense of perceived safety across all items.

The participants were asked to rate how feminine/masculine the avatar’s appearance, voice, and way of speaking were on a seven-point scale. The most feminine condition was assigned with 0 points, the most masculine with 6 points, and neutral with 3 points. Furthermore, to assess how much the appearance, voice, and way of speaking matched the participant’s gender, a gender match index was used. To calculate the match index, the scores for females were first reversed, and this score was then divided by 6 to create an index showing a proportion from 0 to 1. A match index of 0 indicates that the participant’s gender and the perceived gender of the avatar were different, while a match index of 1 indicates that the participant’s gender and the perceived gender of the avatar were the same. Participants were also asked to provide a free-form evaluation of their overall impressions of the interview and specifically about their interactions with the avatar.

### 2.5. Data Analysis

First, descriptive statistics for the pre- and post-session measurements were calculated to provide a summary of participant characteristics and avatar counseling experiences. In addition, Kendall’s correlation coefficients were computed to assess the relationships between variables, and their corresponding significance probabilities were calculated. Owing to the small sample size of this study, correlations of continuous variables were also examined using Kendall’s method, which is more robust against outliers. The significance level was set at 5%. A sample size of 25 shows 80% statistical power for correlations of about 0.6 at a 5% significance level in Kendall’s correlation analysis [69].

## 3. Ethical Considerations

Written informed consent was obtained from all the participants. This study was conducted with the approval of the Ethics Committee of the Tokushima University Hospital (Approval Number: 3650).

## 4. Results

### 4.1. Participant Characteristics

Descriptive statistics for the measurement indicators are shown in Table 1. The participants consisted of six females and nineteen males, with 60% being engineering students. The age range was 21 to 29 years old, with 21- and 22-year-olds being the most common at 36% and 32%, respectively. All participants answered “rarely” for the experience with streaming as an avatar themselves. Only one participant had experienced conversing with an avatar a few times. For viewing avatar-based streams, “rarely” was the most common answer at 40%, but 28% answered “a few times” and 12% answered “about once or twice a week” and “more than three times a week”, indicating that some participants had a certain level of experience. Regarding the Big Five, most scores were between 7 and 8 points, but agreeableness was high at 10.04 points, and conscientiousness was somewhat low at 6.68 points. Autistic tendencies were generally low, similar to those of typical students. The average score for GAD symptoms was 3.3 points, which was considered to be at a level with no symptoms. For social anxiety, the total score was in the borderline range, with social interaction being slightly higher. Scores for both depressive symptoms and suicidal ideation were low.

From the post-measurement results, the average mood during the interview, out of a maximum of 10 points, was highest for happiness at 7.04, followed by anxiety at 3.28, sadness at 1.20, and discomfort at 0.84. The impression of the avatar was generally favorable, with the average per item being 4.06 for likeability, 3.93 for perceived safety, 3.90 for perceived intelligence, and 3.72 for animacy. Anthropomorphism was somewhat lower compared with the others, with an average of 3.33 per item. Regarding the perception of the avatar’s gender, overall, appearance was more masculine at 4, whereas voice and speech were more feminine at 2.20 and 2.80, respectively. For voice, the evaluation by females was particularly feminine at 1.67. Consequently, the gender match index was high for males for appearance at 0.66, high for females for voice at 0.72, and slightly high for females for speech at 0.53.

### 4.2. Topics Discussed during the Interview

The topics discussed during the interview are shown in Table 2. Various contents were discussed, ranging from recent events to events at the time of university entrance and after enrollment several years ago, as well as stories from elementary, junior high, and high school days. For happy events, participants talked about the sense of security from returning to their hometown, successful experiences in sports, games, and academics, satisfaction from travel and shopping, and good relationships with family and friends. For sad events, participants talked about the death or illness of family members or pets, misunderstandings or separations from family and friends, and failures or worries in academics and personal life. For events that made them angry, participants talked about frustrations related to academics and games, immoral acts of others, relationships with parents and friends, and inconveniences in university life.

### 4.3. Correlations between Variables

Significant correlations were found between several variables (Table 3). First, being a technology major was negatively correlated with age and positively correlated with a sad mood during the interview, evaluating voice as masculine, and evaluating speech as masculine. Being a female participant was positively correlated with GS-perceived safety. Experience watching avatar-based streams was positively correlated with neuroticism, GAD, and LSAS.

Extraversion was negatively correlated with AQ social and LSAS, while neuroticism was positively correlated with AQ communication, LSAS total score, and PHQ. AQ communication was positively correlated with SIS. LSAS social situations was positively correlated with feeling anxious during the interview with the avatar, and GS likeability and intelligence were positively correlated with feeling happy during the interview.

Being female was negatively correlated with the match rate between the perceived gender of the avatar’s appearance and the participant’s gender but positively correlated with the perceived gender of the avatar’s voice. Perceiving appearance as masculine was positively correlated with the match rate between the perceived gender of the avatar’s appearance and the participant’s gender and with the match rate between the perceived gender of the avatar’s speech and the participant’s gender. Those who perceived the voice as masculine tended to have a match between their gender and the perceived gender of the avatar’s voice, while those who perceived speech as masculine tended to have a match between their gender and the perceived gender of the avatar’s speech. In addition, there was a positive correlation between the match rate of the avatar’s voice gender with the participant’s gender and the match rate of the avatar’s speech gender with the participant’s gender.

There was a negative correlation between conscientiousness and the match rate between the gender of the avatar’s voice and the gender of the participant’s voice. Positive correlations were observed between anthropomorphism and animacy and likeability, between animacy and likeability, and between likeability and intelligence.

### 4.4. Free-Form Evaluation of the Interview

The participants provided many positive comments about their conversations with the avatar counselor (Table 4). They found the avatars easy to talk to, with appropriate empathy and praise, and found the sessions smooth and enjoyable. The avatar’s gentle voice, demeanor, and active listening skills made even participants who were not comfortable with face-to-face conversations feel at ease. The avatar’s facial expressions, especially its smile, were clear and helped participants feel relaxed and engaged in the conversation. Some participants felt that the experience was meaningful because it provided an opportunity to discuss topics that they would not normally talk about.

However, the participants also reported some negative aspects of the avatar counseling experience (Table 4). Some found it difficult to hear the avatar’s voice, whereas others did not know where the avatar was looking and felt awkward talking to it. Unnatural events, such as the avatar changing clothes in the middle of a session or occasional interruptions in the conversation, were unsettling for some participants. The avatar’s eye movements and facial expressions were not always smooth, making the interaction feel unnatural. Some participants suggested that more detailed facial expressions and quicker responses could improve the naturalness of the conversation. In addition, some participants became anxious about the person behind the avatar and found it difficult to answer abstract questions.

## 5. Discussion

In this study, university students were experimentally interviewed by avatars operated by human counselors and asked to evaluate their experiences. Several interesting findings were obtained from various measurements of participant characteristics, mood during the interview, and impressions of the avatar. In addition, mostly positive opinions about avatar counseling were obtained from the free-form feedback. Meanwhile, the fact that several issues were identified, particularly regarding technical aspects, can be considered points to be considered when conducting future avatar counseling practices and related research.

### 5.1. Mood during the Interview

First, the factors found to be related to the high level of happiness during the interview were the avatar’s impressions of intelligence and likeability (τ = 0.49 and τ = 0.44, respectively). If the communication can evoke positive emotions, positive impressions of the avatar can be promoted, such as intelligence and likeability. Conversely, if the avatar’s appearance and attitude that enhance intelligence and likeability become clear, positive emotions such as happiness may be promoted as well. From the qualitative feedback, participants had a favorable impression of the avatar’s smile and empathetic active listening. Existing research reported that an avatar’s anthropomorphism and likeability lead to credibility [70,71]. This study’s important finding is the connection between intelligence, likeability, and positive emotions during the interview, apart from anthropomorphism, in avatar counseling. Although not quantitatively captured in this study, what may be connecting these factors could be trust and credibility based on the counselor’s expertise.

This study did not quantify the avatar’s smile, but it was well-received according to participant feedback. Meanwhile, while smiles increase the evaluation of a computer agent’s empathy, they reportedly decrease the evaluation of an avatar’s empathy [72]. Because a counselor’s smile is considered essential as an expression of an accepting attitude in human counseling, the results of this study can be said to be more rational. It is thought that a smile leads to a positive evaluation of an avatar only when it is natural and human-like, combined with other information such as vocalization and speech content, not simply because the avatar can express a smile.

In this study, a relationship was found between anxiousness during the interview and LSAS social situations (τ = 0.47). Herein, the topics of conversation were things that made participants happy, sad, or angry, so there were no topics that directly evoked anxiety. Thus, people who are prone to feeling anxious in social situations are thought to also be prone to feeling anxious when communicating with an avatar counselor. From the qualitative feedback, the factors that seemed to cause anxiety included the avatar’s facial expressions being unclear, not knowing what the avatar was doing during moments of silence, and not knowing who the person behind the avatar was. Meanwhile, there was also an opinion that although looking at facial expressions was difficult when communicating with people, it was possible to do so with an avatar, suggesting that counseling with an avatar is not necessarily a bad experience for those with SAD tendencies. Regarding avatar-based therapy for SAD in VR, anxiety about technology can hinder treatment initiation [24]. In avatar counseling using video conferencing systems, especially when targeting clients with high social anxiety, the technology must be carefully explained and anxieties unrelated to therapy must be dispelled.

In addition, regarding mood during the interview, an association was found between being a technology major and sadness. In this study, as a research design, a specific counselor operated the avatar for technology majors, whereas other counselors operated the avatar for medical majors. Therefore, it may not be because they were technology majors that sadness was stronger during the interview; instead, the methods of the counselor in charge of technology majors might have influenced the ease of evoking sadness.

### 5.2. Impressions of the Avatar Counselor

Regarding the Godspeed questionnaire, correlations were found between anthropomorphism, animacy, and likeability (τ = 0.54, τ = 0.50, and τ = 0.49, respectively), as well as between likeability and intelligence (τ = 0.45). The qualitative feedback also suggests that the avatar’s smile and the counselor’s professional responses contributed to the avatar’s naturalness, human likeness, and an atmosphere that made participants feel comfortable talking. The relationship between anthropomorphism and likeability has also been reported in a study where American university students evaluated still images of avatars [70,71], and consistent results with previous research were obtained in this study.

Exaggerated facial expressions and realistic avatars may display uncanniness. In an experiment that exaggerated the facial movements of an animated avatar, authenticity and naturalness decreased as facial expressions became richer, and this relationship was enhanced by excessive empathy [73], Moreover, hyper-realistic avatars and animated avatars induced uncanniness [74]. Meanwhile, in this study, although facial expressions were exaggerated, anthropomorphism was not inhibited. The animated avatar is thought to have prevented excessive realism, and the counselor’s moderate empathy may have contributed to the avatar’s human likeness and naturalness.

In addition, in this study, the avatar’s facial expressions were controlled by motion capture. In social interactions in VR using motion capture, higher levels of facial and body expressions reportedly increase the perception of realism and the quality of interaction [75]. The results of this study, combined with the findings of previous research, suggest that controlling facial expressions through motion capture in video conferencing systems may also enhance anthropomorphism.

Research has reported that anthropomorphism influences avatar credibility and preference [76,77]. Although credibility was not assessed in this study, the relationship between anthropomorphism and likeability as well as that between likeability and intelligence indicate that they reflect credibility and trust in the avatar counselor.

Perceived safety was higher in women. Women may tend to rate the safety of avatars more highly. However, research on robots has reported that women are more prone to feeling anxious or uncomfortable about robots’ unpredictable behavior and negative feedback compared with men [78]. Women also reportedly have higher trust and perceived trust in security robots compared with men [79], showing higher expectations for robots’ agency capabilities [80], and therefore have concerns about the progress of human replacement by robots and the increasing autonomy of intelligent machines [81]. Women may be more prone to feel anxious about having their status threatened or receiving negative feedback because they rate the robots’ performance more highly than men. Meanwhile, avatars are human-operated and do not autonomously cause harm, and women may perceive them as safer. Although larger-scale studies are needed to determine gender differences in this regard, women may be more accepting of avatar counseling.

### 5.3. Perception of Avatar’s Appearance, Voice, and Speech Gender

Regarding the match between the participant’s gender and the perceived gender of the avatar’s appearance, voice, and speech, a negative correlation was observed between conscientiousness and the match in voice (τ = −0.41). As the match between the participant’s and avatar’s voices unlikely influences the participant’s personality, high conscientiousness may decrease the match between the participant’s gender and the perceived gender of the avatar’s voice. Several existing reports have studied the relationship between auditory perception and conscientiousness. For instance, higher conscientiousness and openness are reportedly associated with a lower risk of hearing impairment [82] and a higher acceptable noise level [83]. People with higher conscientiousness may be more sensitive to auditory cues, whereas those with lower conscientiousness may intuitively judge the avatar’s voice as the same gender as themselves. In addition, concerning avatars, people with high conscientiousness are reportedly more likely to perceive their self-created avatars as similar to themselves [84]. Although the current study showed that people with high conscientiousness are more likely to perceive the voice of another person’s avatar as different from their own gender. These two findings might be considered as the two sides of the same coin, reflecting the tendency of people with high conscientiousness. Thus, highly conscientious people are likely to perceive their own avatar as similar to themselves and to perceive others’ avatars as different from themselves. However, even if the participant’s conscientiousness was high in this study, there was no tendency to perceive the appearance or speech of another person’s avatar as different from one’s own gender. Although this study is significant in showing the relationship between user conscientiousness and gender perception of another person’s avatar’s voice, further investigation is required to determine why this relationship was not observed for appearance and speech.

Herein, because the avatar had a male appearance and a female voice, a relationship was observed in females where the match between the avatar’s appearance and their gender decreased (τ = −0.52) and the match between the avatar’s voice and their gender increased (τ = 0.43). Meanwhile, no such relationship was observed for speech (τ = 0.11, n.s.). Even if the vocal tone can be changed using a voice changer, the speech style may not necessarily be perceived to be having the same gender as that of the vocal tone. However, for appearance, voice, and speech, those who perceived the avatar’s gender as male had a higher match between their gender and the perceived gender of the avatar (τ = 0.58, τ = 0.46, and τ = 0.63, respectively). In particular, for males, those who perceived the voice as masculine also perceived the speech as masculine. The gender perception of another person’s avatar’s voice and speech may differ depending on the user’s gender.

The results of this study did not show a relationship between the match of the perceived gender of the counselor avatar’s appearance, voice, and speech with the participant’s gender and the communication experience with an avatar counselor. Users may sometimes prefer to project a specific gender perception onto an androgynous avatar [34], or—in certain contexts—avatars of a specific gender may be effective [33,35]. Moreover, male and female users’ preferences reportedly may differ depending on the agent’s attitude [36]. As this study had a small number of female participants, the impact of user gender and counselor avatar gender on counseling effectiveness must be examined in larger-scale studies.

### 5.4. Influence of Experience with Avatar-Mediated Communication

A positive correlation was found between the experience of watching avatar-based streams and neuroticism, generalized anxiety, and social anxiety (τ = 0.43, τ = 0.42, τ = 0.52, respectively). Viewing certain videos, such as those containing aggression, violence, or drug use, was shown to potentially negatively impact the mental health of youth and adults [85,86]. Although many avatar-based streams are believed to not contain such content, it cannot be said with certainty that there are no harmful ones. In addition, the use of social media may also pose a risk to the mental health of young people [87]. Avatar-based streams, unlike movies and other videos, have a large social media element as the relationship with the streamer is one of the attractions. They may also be conducted in conjunction with actual social networking services. For these reasons, one possible interpretation is that viewing avatar-based streams may be causing mental health issues.

Meanwhile, online media content is not necessarily harmful. For example, suicide-related content may provide an opportunity to openly discuss suicide and lead to help-seeking [88]. Moreover, although the impact of game-related streams is unknown, playing certain video games has been suggested to activate the brain [89]. Previous research also reported that online social support, although not as much as support from real-life friends, may be beneficial to the mental health of adolescents [90]. On the basis of these findings, the relationship found in this study between viewing avatar-based streams and neuroticism, generalized anxiety, and social anxiety may also be due to these participants seeking to view streams with positive intentions, such as improving their mental health or quality of life. In fact, there is some evidence that people with social anxiety find value in interacting with others online [91]. This study did not find any unique experiences of conversing with an avatar counselor among this group of participants. However, as these individuals may benefit from avatar counseling, further investigation is necessary.

### 5.5. Consideration of Other Correlation Analysis Results

A negative correlation was found between being a technology major and age (τ = −0.65), which was thought to reflect the fact that the technology majors were younger than the medical majors. In addition, negative correlations were found between extraversion and AQ social skills and social anxiety (τ = −0.46 and τ = −0.50, respectively), whereas positive correlations were found between neuroticism and AQ communication, social anxiety, and depression (τ = 0.48, τ = 0.44, and τ = 0.42, respectively), and between AQ communication and suicidal ideation (τ = 0.42). These can be said to be conceptually reasonable associations in general. This study did not find any experiences in communication with avatar counselors related to age or these personality and mental state factors.

### 5.6. Participant Feedback

The qualitative feedback from participants was mostly positive, with opinions that the avatar was easier to talk to than initially imagined, that they were able to discuss topics they would not normally talk about, and that it was a meaningful experience. However, there were also some negative comments, particularly regarding technical aspects.

First, regarding the difficulty in hearing the voice, this could be attributed to the limited microphone sensitivity due to the use of a voice changer in this study. Voice changers pick up noise and emit some kind of sound, so the counselor operating the avatar must either be in a completely quiet space and avoid making any noise or limit the microphone sensitivity. In practice, clients may also be required to use earphones as a countermeasure.

Next, the avatar’s clothing changing in the middle of the interview was due to misoperation as the avatar used in this study allowed for clothing changes and motions to be presented through button operations. Thus, counselors operating avatars must be required to have a certain level of skill in avatar operation.

The conversation stopping midway or the avatar disappearing (thus causing distress) is thought to be due to the influence of the communication environment and terminal performance. In addition to the web conferencing software, multiple software programs must be run simultaneously, including software that captures the counselor’s facial expressions from the camera to operate the avatar, software that imports the avatar image and sends it to the web conferencing system, and voice changer software. Therefore, the communication environment and terminal performance must be thoroughly checked in advance.

The pointing out of the slowness of responses and reactions is also thought to be influenced by terminal performance to a certain extent. In addition, voice changers cause a certain amount of delay, so a system selection that considers the merits and demerits of voice changing is required.

Finally, comments about the avatar’s eye movements and facial expression changes not being smooth or that more detailed facial expression changes would be better are issues related to avatar development. This study used a commercially available avatar; therefore, the development of avatars that can express facial expression changes in more detail for counseling purposes may be necessary.

### 5.7. Limitations and Future Research Challenges

Although this study provides important insights on the provision of counseling through avatars, it has several limitations. First, this was a preliminary exploratory study, with a small sample size and no control group. Therefore, we may have missed sub-moderate (smaller than τ = 0.6) correlations. In addition, we have not yet quantitatively investigated the effects of avatar counseling. These points need to be explored in more rigorous design studies.

Second, regarding the lack of association between prior experience communicating with others embodied as avatars and various variables, most participants answered “rarely” for their experience communicating with others who are embodied as avatars, and only one participant answered “a few times”. In addition, no participants had experience communicating with others while they themselves were embodied as avatars. The experience of conversing with an avatar counselor of those who already have experience communicating through avatars must be examined in larger-scale studies.

Third, regarding the correlation between the experience of viewing avatar-based streams and neuroticism, generalized anxiety, and social anxiety, as well as the correlation between neuroticism and AQ communication, social anxiety, and depression, no unique experiences of conversing with avatar counselors were found for these characteristics. In the future, it is necessary to further examine how avatar counseling can produce positive effects, particularly for individuals with tendencies toward mental health issues such as neuroticism, generalized anxiety, and social anxiety.

Fourth, while it was suggested that people with high social anxiety experienced anxiety during the interview, how this affected the effectiveness of the counseling session is unclear. Anxiety may also be evoked in interviews with human counselors, and a certain level of anxiety evocation may be necessary for effective counseling. However, it is also true that there is a unique cause of anxiety during counseling with avatars: not knowing who is operating the avatar. These aspects must be examined thoroughly from the perspective of the advantages and disadvantages of counseling with avatars, particularly for those with high social anxiety.

Fifth, the study did not allow for determining the directionality of causality regarding the relationship between the high level of happiness during the interview and the impressions of intelligence and likeability toward the avatar. Further verification is needed to determine whether positive emotions during the interview lead to favorable impressions of the avatar or whether favorable impressions of the avatar lead to the promotion of positive emotions during the interview. Furthermore, anthropomorphism was not related to happiness or intelligence. However, correlations were observed between likeability, intelligence, and happiness as well as between anthropomorphism and animacy and likeability. These variables are thought to be also related to credibility and trust; however, we were unable to assess them. How these factors lead to high evaluations of the avatar and interview and how this affects the effectiveness of counseling are important issues to be considered in the future.

Finally, this study suggested the possibility of the avatar’s smile being related to positive impressions of the avatar and positive emotional responses during the interview. However, the avatar’s smile was not quantified. Future research must quantify avatar’s smiles, verify the above relationships, and elucidate what kind of smile is more effective.

## 6. Conclusions

This study experimentally examined university students’ conversations with counselors through avatars and obtained quantitative and qualitative results, which showed that positive emotions during the interview were associated with the impressions of the avatar’s intelligence and likeability. The study also revealed interrelationships among the avatar’s impressions of anthropomorphism, animacy, likeability, and intelligence, indicating that factors such as the avatar’s smile and the counselor’s expertise in empathy and approval may have contributed to these impressions. The findings indicated that high conscientiousness might increase the match between the participant’s gender and the perceived gender of the avatar’s voice. Perceptions of the avatar’s speech style may vary more among participants than perceptions of the avatar’s appearance and voice gender. However, no clear influence of the match between the participant’s gender and the perceived gender from the avatar’s appearance, voice, and speech style on communication with the avatar was observed. A relationship was observed between the experience of watching avatar-based streams and neuroticism, generalized anxiety, and social anxiety, although no specific experiences related to counseling with avatars were confirmed concerning these factors. Anxiety during the interview was observed to be related to social anxiety. Accordingly, implications for practice and future research issues are provided in Table 5. Practical knowledge and empirical research on avatar-based counseling are anticipated to expand in the future, including investigations of long-term effectiveness and comparisons with online counseling using real images.

## Figures and Tables

**Figure 1 healthcare-12-01287-f001:**
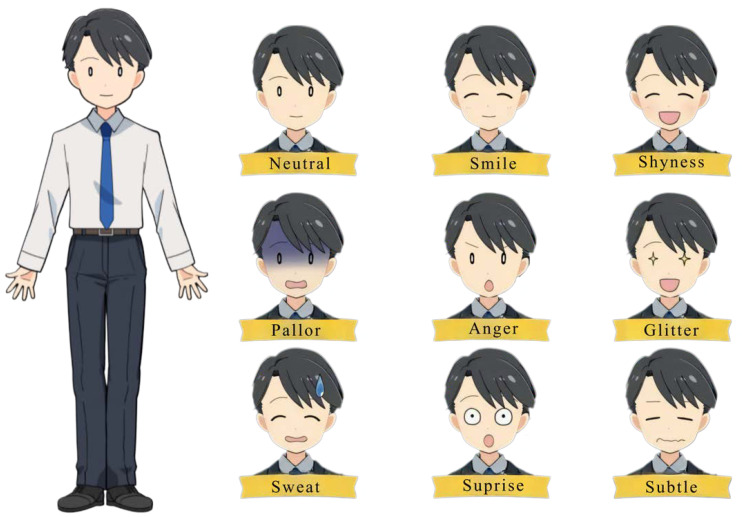
Avatar appearance.

**Figure 2 healthcare-12-01287-f002:**
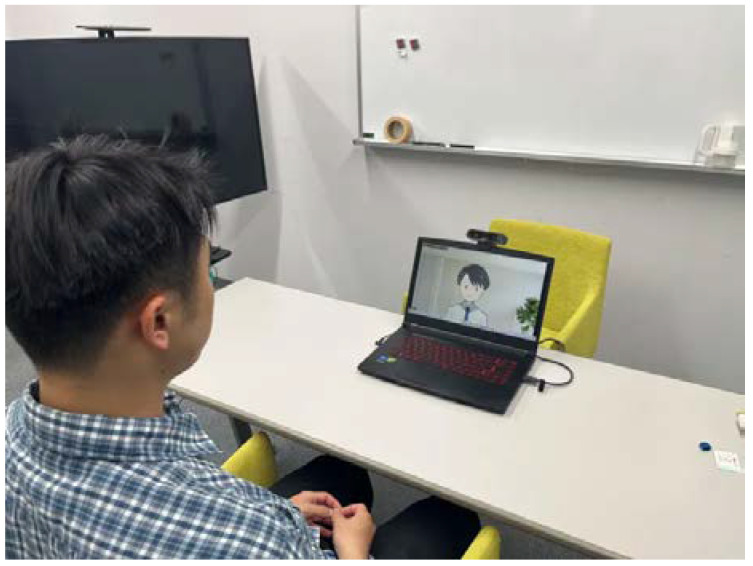
Scene of the experiment reproduced by the author.

**Figure 3 healthcare-12-01287-f003:**
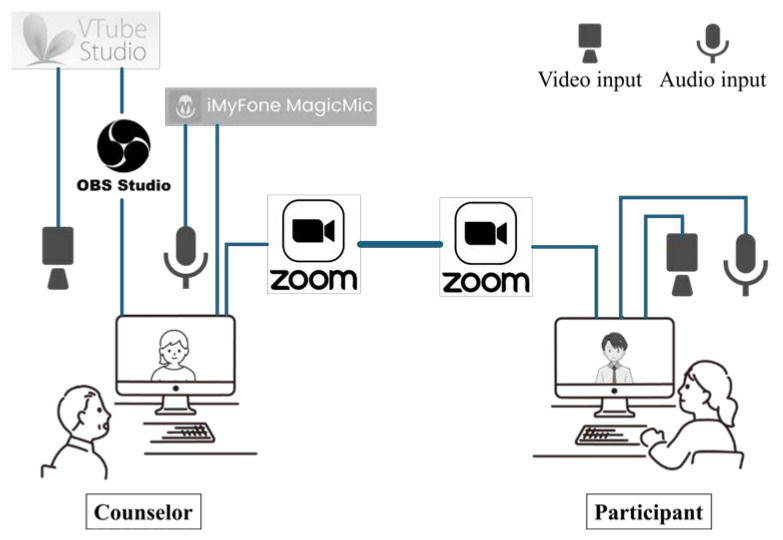
Overview of the experiment system. The counselor’s video input was processed using VTube Studio, which performed facial expression tracking and avatar control. The resulting video output was then fed into OBS Studio, where it was captured and sent to Zoom as a video feed. The counselor’s audio was processed through MagicMic, which transformed the voice before inputting it into Zoom. On the client’s side, the video and microphone inputs were left unaltered and fed directly into Zoom, preserving their original form.

**Table 1 healthcare-12-01287-t001:** Participant characteristics.

	Overall, *N* = 25	Women, *N* = 6	Men, *N* = 19
Major			
Medicine	10 (40%)	3 (50%)	7 (36.84%)
Technology	15 (60%)	3 (50%)	12 (63.16%)
Age			
21	9 (36%)	3 (50%)	6 (31.58%)
22	8 (32%)	0 (0%)	8 (42.11%)
23	2 (8%)	0 (0%)	2 (10.53%)
24	2 (8%)	1 (16.67%)	1 (5.26%)
25	1 (4%)	0 (0%)	1 (5.26%)
27	2 (8%)	2 (33.33%)	0 (0%)
29	1 (4%)	0 (0%)	1 (5.26%)
Avatar experience			
Self			
Almost never	25 (100%)	6 (100%)	19 (100%)
Other			
Almost never	24 (96%)	6 (100%)	18 (94.74%)
A few times	1 (4%)	0 (0%)	1 (5.26%)
Watch			
Almost never	10 (40%)	2 (33.33%)	8 (42.11%)
A few times	7 (28%)	2 (33.33%)	5 (26.32%)
Once a month	2 (8%)	1 (16.67%)	1 (5.26%)
One or two times a week	3 (12%)	1 (16.67%)	2 (10.53%)
More than three times a week	3 (12%)	0 (0%)	3 (15.79%)
Individual characteristics			
TIPI			
Extraversion	7.60 ± 2.38	6.67 ± 3.08	7.89 ± 2.13
Agreeableness	10.04 ± 1.90	10 ± 2	10.05 ± 1.93
Conscientiousness	6.68 ± 2.29	5.17 ± 1.60	7.16 ± 2.29
Neuroticism	7.80 ± 2.06	8.33 ± 1.37	7.63 ± 2.24
Openness	8.56 ± 1.94	8.33 ± 2.66	8.63 ± 1.74
AQ			
Total	19.76 ± 5.84	17.67 ± 7.12	20.42 ± 5.43
Social skill	3.84 ± 2.93	4 ± 3.79	3.79 ± 2.72
Attention shifting	4.76 ± 1.67	4.83 ± 1.33	4.74 ± 1.79
Attention to detail	4.68 ± 1.70	3.50 ± 1.22	5.05 ± 1.68
Communication	3.16 ± 2.01	3.33 ± 3.01	3.11 ± 1.70
Imagination	3.32 ± 1.75	2 ± 1.41	3.74 ± 1.66
GAD-7	3.28 ± 3.47	2 ± 1.79	3.68 ± 3.80
LSAS			
Total	40.44 ± 21.74	50 ± 19.15	37.42 ± 22.10
Performance	18.16 ± 10.36	21.17 ± 9.70	17.21 ± 10.63
Social interaction	22.28 ± 11.98	28.83 ± 9.81	20.21 ± 12.07
PHQ-9	3.68 ± 4	2.33 ± 1.97	4.11 ± 4.41
SIS	1.32 ± 1.95	1.33 ± 1.63	1.32 ± 2.08
Emotions during interview			
Happiness	7.04 ± 1.93	7.83 ± 1.33	6.79 ± 2.04
Anxiety	3.28 ± 2.44	4 ± 3.03	3.05 ± 2.27
Discomfort	0.84 ± 0.94	1 ± 0.89	0.79 ± 0.98
Sadness	1.20 ± 1.91	0.33 ± 0.82	1.47 ± 2.09
GodSpeed: Avatar impression			
Anthropomorphism	16.64 ± 3.38	15 ± 1.55	17.16 ± 3.66
Animacy	22.32 ± 3.91	21.50 ± 1.64	22.58 ± 4.40
Likeability	20.32 ± 3.16	20.83 ± 2.04	20.16 ± 3.47
Perceived intelligence	19.52 ± 2.60	20.33 ± 3.44	19.26 ± 2.33
Perceived safety	11.80 ± 2.02	13.33 ± 1.03	11.32 ± 2.03
Perceived avatar gender			
Appearance (men)	4 ± 1.29	4.17 ± 0.75	3.95 ± 1.43
Voice (men)	2.20 ± 1.58	1.67 ± 1.51	2.37 ± 1.61
Speech (men)	2.80 ± 1.35	2.83 ± 1.17	2.79 ± 1.44
GENDER MATCH INDEX			
Appearance	0.57 ± 0.26	0.31 ± 0.12	0.66 ± 0.24
Voice	0.47 ± 0.29	0.72 ± 0.25	0.40 ± 0.27
Speech	0.48 ± 0.23	0.53 ± 0.20	0.47 ± 0.24

Avatar experience: “Self” refers to the experience of communicating with an avatar by oneself, “Other” refers to the experience of communicating with an avatar by another person, and “Watch” refers to the experience of watching a streaming video by an avatar. TIPI: Ten-Item Personality Inventory. AQ: Autism spectrum quotient. GAD-7: Generalized Anxiety Disorder-7. LSAS: Liebowitz Social Anxiety Scale. PHQ-9: Patient Health Questionnaire-9. SIS: Suicidal Ideation Scale. GENDER MATCH INDEX takes a value between 0 and 1, and is 1 if the participant’s gender matches the participant’s perceived avatar gender.

**Table 2 healthcare-12-01287-t002:** Topics discussed during the interview.

Category	Experiences
Happy experiences	- Joy from adapting well to environmental changes associated with entering university or graduate school - Good experiences from travel or dining - Achievements in sports clubs during middle and high school - Memorable experiences of buying special items like cosmetics or a car as an adult - Experiences of feeling connected with family and relatives - Success in academics or entrance exams - Achieving goals or victories in competitive sports
Sad experiences	- Illness or death of family members or pets - Loss of relationships, such as breakups with partners or changes in friendships - Failures or setbacks in school or personal life - Social issues, such as economic concerns due to earthquakes or rising prices - Feelings of loneliness or isolation in personal life
Angry experiences	- Frustration with one’s own lack of skill or poor academic performance - Anger toward friends’ actions, such as breaking promises or speaking ill of you - Anger toward others’ poor driving manners or rude customers at part-time jobs - Anger toward others’ malice, such as verbal abuse online or tampering with personal belongings - Anger toward environments or situations, such as changes in online class schedules or not winning at gambling - Unpleasant childhood experiences, such as fights with friends or gossip

**Table 3 healthcare-12-01287-t003:** Kendall’s correlation coefficient between variables (excerpts).

		1	2	3	6	7	9	10	12	13	16	19	20	21	24	28	29	30	32	34	35	36
1	age	-																				
2	gender: women	0.06	-																			
3	major: technology	−0.65 **	−0.11	-																		
6	avatar: watch	−0.07	0.01	0.23	-																	
7	TIP: extraversion	0.02	−0.21	−0.14	−0.26	-																
9	TIP: conscientiousness	−0.28	−0.37	0.10	−0.12	−0.01	−															
10	TIP: neuroticism	−0.01	0.09	0.35	0.43 *	−0.36	−0.23	−														
12	AQ: total	0.10	−0.14	0.10	0.16	−0.39	0.11	0.32	−													
13	AQ: social	0.12	−0.01	−0.04	0.05	−0.46 *	0.17	0.13	0.56 **	−												
14	AQ: shifting	0.05	0.05	0.11	0.12	−0.15	0.16	0.25	0.49 *	0.42 *												
16	AQ: communication	0.14	−0.02	0.09	0.33	−0.37	−0.14	0.48 *	0.68 **	0.41 *	−											
18	GAD-7	0.16	−0.12	0.09	0.42 *	−0.11	0.01	0.31	0.25	0.12	0.19											
19	LSAS: total	0.06	0.23	0.10	0.52 **	−0.50 *	−0.14	0.44 *	0.21	0.24	0.35	−										
20	LSAS: performance	−0.03	0.16	0.21	0.55 **	−0.44 *	−0.14	0.50	0.23	0.2	0.39	0.87 **	−									
21	LSAS: social	0.05	0.26	0.03	0.52 **	−0.53 **	−0.12	0.37	0.19	0.26	0.28	0.89 **	0.75 **	−								
22	PHQ-9	−0.08	−0.11	0.38	0.27	−0.09	−0.20	0.42 *	0.29	0.09	0.29	0.27	0.31	0.23								
23	SIS	0	0.06	0.05	0.31	−0.31	−0.04	0.27	0.33	0.22	0.42 *	0.12	0.11	0.16								
24	happiness	0.32	0.19	−0.24	0.07	−0.28	−0.31	0.04	0.13	0.02	0.17	0.13	0.09	0.17	−							
25	anxiety	−0.07	0.13	0.02	0.39	−0.14	0.14	0.10	0.04	0.21	−0.01	0.39	0.36	0.47 *	−0.17							
27	sadness	−0.21	−0.28	0.55 **	0.39	−0.14	0.07	0.31	0.16	0.11	0.08	0.23	0.26	0.2	−0.19							
29	voice: male	−0.22	−0.18	0.54 **	0.13	0.08	0.05	0.10	−0.02	−0.17	−0.02	−0.08	−0.01	−0.15	−0.05	0.20	−					
30	speech: male	−0.18	0.02	0.50 *	0.12	−0.06	0.10	0.12	−0.01	−0.12	−0.07	−0.03	0.04	−0.11	0.02	0.34	0.69 **	−				
31	appearance: match	−0.19	−0.52 **	0.35	0.06	0.02	0.28	0.02	0.19	0.03	0.01	−0.23	−0.17	−0.26	0.02	0.58 **	0.35	0.35				
32	voice: match	0.09	0.43 *	0.16	−0.07	−0.04	−0.41 *	0.16	−0.07	−0.08	0.06	−0.09	−0.12	−0.13	0.22	0.32	0.46 *	0.39	−			
33	speech: match	−0.01	0.11	0.32	−0.02	−0.16	−0.12	0.18	0.2	0.12	0.18	−0.13	−0.08	−0.17	0.15	0.45 *	0.39	0.63 **	0.67 **			
34	GS: anthropomorphism	−0.05	−0.24	−0.11	0.04	−0.04	0.26	0.03	0.11	0	−0.08	−0.04	−0.06	0.03	0.18	0.14	−0.03	−0.08	−0.23	−		
35	GS: animacy	−0.04	−0.15	0.02	−0.10	−0.07	0.18	−0.03	0.14	−0.01	−0.13	−0.1	−0.12	−0.05	0.17	0.27	0.08	0.07	−0.13	0.54 **	−	
36	GS: likeability	0.11	0.05	−0.24	0.07	−0.09	−0.09	0.02	0.11	−0.13	0.02	0.07	0.01	0.11	0.44 *	0.22	−0.08	−0.02	−0.07	0.50 *	0.49 *	−
37	GS: intelligence	0.02	0.13	0	0.09	−0.31	−0.25	0.10	0.17	−0.01	0.18	0.16	0.14	0.19	0.49 *	−0.04	0.01	−0.05	0.07	0.28	0.29	0.45 *
38	GS: safety	0.04	0.43 *	−0.14	0.02	−0.20	−0.24	−0.06	−0.10	−0.02	−0.05	0.24	0.20	0.27	0.07	−0.08	−0.10	−0.07	0.07	−0.19	0.03	0.14

* *p* < 0.05, ** *p* < 0.01. This is the result of excerpting only those variables for which even one significant correlation was observed. Please refer to Appendix A for correlations among all variables. “avatar: watch” means the avatar’s experience of watching others. TIP: Ten-Item Personality Inventory. AQ: autism spectrum quotient. GAD-7: Generalized Anxiety Disorder-7. LSAS: Liebowitz Social Anxiety Scale. PHQ-9: Patient Health Questionnaire-9. SIS: Suicidal Ideation Scale. Happiness, anxiety, and sadness are moods during conversations. “voice: male” indicates the perceived masculinity of the avatar’s voice, and “speech: male” indicates the perceived masculinity of the avatar’s speech. “appearance: match” is the gender match between the participant’s gender and the avatar’s appearance, “voice: match” is the gender match between the participant’s gender and the avatar’s voice, and “speech: match” is the gender match between the participant’s gender and the avatar’s speech. GS: Godspeed.

**Table 4 healthcare-12-01287-t004:** Participant feedback on the avatar counseling.

Category	Opinions
Positive opinions	- The avatar was easy to talk with, with appropriate empathy and praise, making the counseling session smooth. - It was easier to talk with the avatar than initially expected. - Although I struggle with face-to-face conversations, I could see the avatar’s facial expressions. - The avatar’s voice and demeanor were gentle, allowing me to feel at ease during the session. - The avatar’s reactions, active listening, and summarizing skills were good, making the conversation enjoyable. - The avatar’s facial expressions, especially the smiles, were clear, enabling me to relax and converse. - The avatar felt real, natural, and without any sense of discomfort. - The avatar’s smiling while speaking felt human-like. - It was an opportunity to discuss topics I don’t usually talk about, making it a meaningful experience.
Negative opinions	- Sometimes the voice was difficult to hear. - Talking to an avatar felt awkward, and it was hard to tell where the avatar was looking. - The avatar’s clothing change mid-session, which felt unnatural. - The conversation occasionally stopped or the avatar disappeared, which was unsettling. - The avatar’s eye movements (showing the whites of the eyes or unnatural lines) and facial expression changes were not smooth, making it feel unnatural. - More detailed facial expressions would make the conversation easier. - The avatar’s facial expressions were unclear, making it difficult to respond during moments of silence. - Instead of silence when thinking, having a voice expressing contemplation would be better. - Responses could be slightly quicker for a more natural conversation. - There seemed to be a mismatch between the avatar’s appearance and voice. - I felt slightly anxious, wondering who the person behind the avatar was. - There were abstract questions that were difficult to answer. - The counselor’s typical style of encouraging reflection made the conversation a bit challenging.

**Table 5 healthcare-12-01287-t005:** Practical implications and research challenges.

Category	Implications/Challenges
Practical implications	- The avatar’s smile and empathetic active listening may elicit positive emotions from the client and lead to a favorable impression of the avatar. - For clients with high social anxiety, careful explanations about the technology must be provided to reduce their anxiety. - To enhance the human likeness and naturalness of the avatar, balancing a not-too-realistic anime-style design with moderate expressions of empathy by the counselor is crucial. - Motion capture for controlling the avatar’s facial expressions contributes to improving human likeness. - Female clients may be more receptive to avatar counseling. - The perceived gender of the avatar’s voice may be related to the client’s personality traits (conscientiousness). - Counselors must acquire avatar operation skills and strive for smooth communication, paying particular attention to the avatar’s facial expressions, movements, and voice quality. - In avatar counseling, solving technical issues such as audio quality, communication environment, and device performance important. - In the development of avatars for counseling, detailed expressions of facial changes are required.
Research challenges	- Clarifying the differences in experiences with avatar counselor sessions based on prior experience communicating through avatars, by conducting research on individuals with such experience. - Verifying the effectiveness of avatar counseling for individuals with mental health issues, particularly those with extensive experience viewing avatar-delivered content. - Examining in detail the impact of anxiety arousal in avatar counseling for individuals with high social anxiety from the perspective of counseling effectiveness. - Identifying causal relationships between positive emotions during the session and favorable impressions of the avatar. - Elucidating relationships among factors such as the avatar’s anthropomorphism, likeability, intelligence, positive emotions during the session, trust, and credibility and their impact on counseling effectiveness. - Quantifying the avatar’s smile, examining the relationship between smiles and positive impressions or emotional responses, and identifying the characteristics of effective smiles. - Clarifying the impact of the perceived gender of the counselor avatar, the client’s gender, and the degree of match between the perceived gender of the avatar and the client’s gender on counseling effectiveness.

## Data Availability

The data presented in this study are available on request from the corresponding author.

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
