# Peer review of "An Exploratory Study of the Potential of Online Counseling for University Students by a Human-Operated Avatar Counselor"

_healthcare, 2024, doi:10.3390/healthcare12131287_

Round 1

Reviewer 1 Report

Comments and Suggestions for Authors

The work explores the use of online counseling by a human avator counselor. The focus is on university students. A study is conducted in this regard and results are presented. Following are some of the comments that needs to be addressed

1. The Introduction is written well but there are too many subheadings crowded in the Introduction section. This section can be divided into two sections where one is focused on general Introduction and other is focused on literature review part. 

2. The authors must clarify the sample set taken for the study and whether it is enough to make meaningful conclusion. 

3. In Section 2, there are too many subsections which makes section quite hard to follow. I suggest to make less subsections. Either make new section or make less subsections and merge them. 

4. Table 3 size needs to be reduced and should be made more clear. 

5. In Section 4, some numbers from Table 3 can be quoted to support the discussion. 

6. Table 5 must be appropriately placed. I think it can be better added before conclusions.

Comments on the Quality of English Language

Fine.

Author Response

We would like to express our sincere gratitude to Reviewer 1 for the insightful comments and suggestions. The feedback on the structure of the Introduction, sample size, and organization of the Methods section has greatly contributed to improving the clarity and coherence of our manuscript. We also appreciate the suggestions regarding the presentation of Tables 3 and 5, which have enhanced the readability of our results and discussion.

Comments 1: The Introduction is written well but there are too many subheadings crowded in the Introduction section. This section can be divided into two sections where one is focused on general Introduction and other is focused on literature review part. 

Response 1: We agree with reviewer 1’s point. We have added a section to the Introduction called "1.1. Literature review."

Comments 2: The authors must clarify the sample set taken for the study and whether it is enough to make meaningful conclusion. 

Response 2: We agree with the points raised by reviewer 1 and have added the following to the Participants, Statistical Analysis, and Limitations sections. Since this is an exploratory study, no prior sample size calculation was conducted. We recruited as many subjects as possible within our budget. The sample size of 25 provides 80% power at the 5% level for correlations of about 0.6 in Kendall's correlation analysis (May and Looney, 2020). Therefore, correlations smaller than 0.6 may have been overlooked in the analysis of this study.

May, Justine O., and Stephen W. Looney. "Sample size charts for Spearman and Kendall coefficients." Journal of biometrics & biostatistics 11.6 (2020): 1-7.

Comments 3: In Section 2, there are too many subsections which makes section quite hard to follow. I suggest to make less subsections. Either make new section or make less subsections and merge them. 

Response 3: Since this is a methods section, there were many formal subsections, but we reduced the number of subsections wherever possible and made the Ethical Considerations section independent.

Comments 4: Table 3 size needs to be reduced and should be made more clear. 

Response 4: The entirety of Table 3 was moved to the Supplementary Materials. We then excerpted the important parts and composed a new Table 3.

Comments 5: In Section 4, some numbers from Table 3 can be quoted to support the discussion. 

Response 5: Results of data analysis have been added to the discussion as needed.

Comments 6: Table 5 must be appropriately placed. I think it can be better added before conclusions.

Response 6: We have moved Table 5 to before the Conclusion section.

Reviewer 2 Report

Comments and Suggestions for Authors

Thank for you submitting your manuscript on "An Exploratory Study of the Potential of Online Counseling for University Students by a Human-Operated Avatar Counselor." As a reviewer who has a bit of experience in the study of digital avatars and mental health/well-being, your study was an interesting one to read through. My only concern is just some further clarification on some of your methodology:

- Why were the participants asked those three specific questions...is this a common thing done for the counseling field, or was recommended by other mental health professionals, chosen at random? Just was curious as to why these questions were chosen.

- Why was that particular, anime-style avatar chosen...I know it was "free," but I'm sure there's others that were free too--was it because it was androgynous leaning, or some other reason?

- Curious as to why the decision was made to do an elder woman voice, particularly since the anime-style avatar appears to have been chosen because of its youth and androgyny? 

- I noticed that the research was funded by JKA (am assuming Japanese Karate Association?) Is there any association between the research conducted, or was this more of just random funding that was chosen to be put on this particular avatar-based mental health research?

Author Response

We would like to thank Reviewer 2 for the valuable comments and questions. The inquiries about the rationale behind the specific questions asked to participants, the choice of the anime-style avatar, and the use of an elderly woman's voice have helped us clarify important aspects of our study design. We also appreciate the attention to the funding source and its potential association with the research conducted. The feedback has been instrumental in improving the transparency and overall quality of the manuscript.

Comment 1: Why were the participants asked those three specific questions...is this a common thing done for the counseling field, or was recommended by other mental health professionals, chosen at random? Just was curious as to why these questions were chosen.

Response 1: Focusing on happy, sad, and angry emotions is often done in the context of research on emotions (Mitchell et al., 2001; Dechert et al., 2005; Páez et al., 2013). We chose these topics because they were more akin to a counseling context than simple everyday conversation. Additionally, these emotions are common experiences among college students and can be easily discussed, unlike current everyday problems or specific goals they want to achieve. We have added these explanations to the Methods section.

Mitchell, Monique, et al. "The effects of anger, sadness and happiness on persuasive message processing: A test of the negative state relief model." Communication Monographs 68.4 (2001): 347-359.
Dechert, Nathan T., William Flack Jr, and Francis Craig Jr. "Patterns of cardiovascular responses during angry, sad, and happy emotional recall tasks." Cognition and Emotion 19.6 (2005): 941-951.
Páez, Darío, et al. "Affect regulation strategies and perceived emotional adjustment for negative and positive affect: a study on anger, sadness and joy." The Journal of Positive Psychology 8.3 (2013): 249-262.

Comment 2: Why was that particular, anime-style avatar chosen...I know it was "free," but I'm sure there's others that were free too--was it because it was androgynous leaning, or some other reason?

Response 2: For the avatar's appearance, based on the findings of previous studies, we adopted a more human-like, not too realistic, gender-neutral look. In addition, we selected an avatar with a clear expression of a smile. An explanation of this point has been added to the relevant section of the Methods.

Comment 3: Curious as to why the decision was made to do an elder woman voice, particularly since the anime-style avatar appears to have been chosen because of its youth and androgyny?

Response 3: In this study, a voice changer was used to reduce the influence of the human manipulating the avatar. For the voice, we also tried to choose a gender-neutral one. As a result, the most appropriate pitch was one that was neither too high nor too low and was described as "elderly woman" in the system. We have added this description to the Methods section.

Comment 4: I noticed that the research was funded by JKA (am assuming Japanese Karate Association?) Is there any association between the research conducted, or was this more of just random funding that was chosen to be put on this particular avatar-based mental health research?

Response 4: The source of funds, KEIRIN RACE, refers to bicycle racing, and JKA (Japan Keirin Auto Race Foundation) provides research grants as a social contribution using the profits from the bicycle racing business. JKA mainly subsidizes research on machinery broadly derived from bicycles and motorcycles. While not directly related to this study, the research falls under the broad category of machinery.

Reviewer 3 Report

Comments and Suggestions for Authors

The manuscript presents the assessment of online counselling scenario where counsellor’s virtual appearance is presented to the participants. The study of effectiveness of  counselor operated avatars is an important 

research area where this manuscript provides significant contribution by analysing the correlations of various disorders. The overall results look well explained and discussed. There are few comments and suggestions which could be helpful in revision.

  1. How was the number of participants decided? Was any power analysis done? The number of participants looks lower than usual. Would that affect the conclusion?
  2. Section 2.2 could include more details on the participant selection with selection criteria. It is important to ensure the physical as well as mental soundness of the participants. Was any screening done prior tot he experiments to ensure that the observations were purely related to the experiments.
  3. Results and discussion section presents detailed analysis of the variables. However with large number of variables, the description of the correlation values often fails to conclude the observations.  

Author Response

We would like to express our appreciation to Reviewer 3 for the thoughtful comments and questions regarding the sample size, participant selection criteria, and the analysis of the large number of variables. The feedback has been invaluable in addressing the limitations of our study and highlighting the need for a larger sample size to draw more definitive conclusions. We are grateful for the insights, which have helped us to better contextualize our findings and acknowledge the need for further research to explore the influence of gender differences in this context.

Comment 1: How was the number of participants decided? Was any power analysis done? The number of participants looks lower than usual. Would that affect the conclusion?

Response 1: Since this study was exploratory in nature, the number of participants was set to be as large as possible within the limits of the budget. As a result, 25 participants were obtained. We performed a Kendall's correlation analysis, which shows that for 25 participants, at a 5% significance level, 80% power is obtained at τ = 0.6. We have added an explanation for this in the Participants and Statistical Analysis sections, pointing out that correlations with low marginal scores may have been overlooked. This limitation has been noted in the Limitations section.

Comment 2: Section 2.2 could include more details on the participant selection with selection criteria. It is important to ensure the physical as well as mental soundness of the participants. Was any screening done prior tot he experiments to ensure that the observations were purely related to the experiments.

Response 2: Participants were recruited in order of earliest application up to the pre-determined number of openings. Exclusion criteria included not being physically or mentally fit enough to understand the instructions in the study and to tolerate the study; however, none of the participants met these criteria, and all were recruited. We have added this point to the Participants section.

Comment 3: Results and discussion section presents detailed analysis of the variables. However with large number of variables, the description of the correlation values often fails to conclude the observations.  

Response 3: Indeed, there were many variables used in the analysis in this study. However, we are aware that due to the small sample size, there did not seem to be as many significant correlations observed. Furthermore, we intend to prioritize, organize and discuss the results. Certainly, we were limited in our ability to discuss the likely influence of gender differences, especially since the association between the gender of the participants and the perceived gender of the counselor avatar was not clear in some aspects. As mentioned in the Limitations section, this point needs to be tested in a larger study.

Round 2

Reviewer 1 Report

Comments and Suggestions for Authors

Authors have made changes as per comments in the last round.